# The Origin, Application and Mechanism of Therapeutic Climbing: A Narrative Review

**DOI:** 10.3390/ijerph19159696

**Published:** 2022-08-06

**Authors:** Sheng Liu, Xiaoqin Gong, Hanping Li, Yuan Li

**Affiliations:** School of Physical Education, China University of Geosciences, Wuhan 430074, China

**Keywords:** therapeutic climbing, chronic disease, exercise intervention, mechanism

## Abstract

As an innovative exercise therapy, therapeutic climbing (TC) has attracted more attention than ever before in recent years. In this review of the related studies on TC, the authors explore its origin and development; summarize its therapeutic effect in treating depression, low back pain and other diseases; and further analyze its underlying mechanism. According to the literature, TC was primarily applied in the field of orthopedics and then was gradually used in neurology, psychiatry and psychology. It provides a new means for the treatment of depression, lower back pain, multiple sclerosis and other diseases. There are two potential mechanisms: physiological and psychological. In the future, exercise effects, adverse effects and exercise prescriptions of TC should be explored with large samples and high-quality randomized controlled trials.

## 1. Introduction

Based on the notion that exercise is medicine, there is an impetus to integrate physical activities with medical systems of prevention, rehabilitation and treatment to improve public fitness and health [1]. In recent years, more and more people have advocated for physical activity, balneotherapy and music therapy as ways to treat chronic diseases, and they emphasized their key role in clinical non-pharmacological intervention [2,3,4,5,6,7]. According to the theory of evolution, climbing is the most primitive form of human movement. Its uniqueness, strong motivation development and control of movement intensity, together with the development of safety aids and technologies, enable participants to experience rock climbing under safe conditions and overcome a psychological boundary via “self-determination theory”. These factors have made rock climbing a near-ideal “therapeutic sport” [8]. TC is one of the exercise modalities used in the treatment of depression and various disorders, which consists of elements of resistance and whole-body strength endurance training [9,10] and also contains unique psychosocial features [8]. TC takes different forms, for example, bouldering without a rope or lead climbing with the protection of a partner. When participating in TC, doctors and professional instructors will individualize exercise programs based on the results of pre-exercise screenings. TC is not suitable for patients with advanced osteoporosis, extreme obesity, incomplete healing of fractures, acute illness, acute pain, etc. However, the use of TC was researched in fields such as orthopedics, neurology, psychology and psychiatry with positive results [11,12,13,14]. The purpose of this narrative review is to explore the origins and development of TC in order to deeply discuss the effectiveness, limitations and underlying mechanisms of TC for patients with depression, lower back pain, multiple sclerosis and other disorders.

## 2. Methods

In the literature review part, a search was conducted using five scientific literature databases. We searched 617 articles, of which 410 were selected for pre-screening after eliminating duplicates. The selection process of this review is shown in Figure 1. We searched on the five different platforms (Web of Science, PubMed, EBSCO, Cochrane Library, Google Scholar) by using the following query terms: therapeutic climbing, rock climbing, sport climbing, speed climbing, lead climbing, bouldering, fitness, chronic diseases, benefit and health. Furthermore, we checked the References lists to confirm the relative studies. To refine the scope of our narrative review, we established a set of inclusion criteria. According to the criteria, articles published up to December 2021 in English or German as the main topic of TC were included, such as original scientific papers, academic dissertation, clinical trials and meta-analyses. The exclusion criteria included the full text not being available, incomplete data, conference papers and letters. Pre-screening is based on the title and abstract of the paper to exclude irrelevant papers. Based on these criteria, we identified 119 papers that focused specifically on the field of TC. Then, full-text papers were reviewed for their relevance to TC. So, 84 full-text articles of these papers were searched and fully reviewed, and 79 of them met our defined inclusion and exclusion criteria.

## 3. The Origin and Development of TC

Physiologically speaking, TC can strengthen one’s overall physical quality by increasing muscle strength and endurance, as well as body flexibility, coordination and balance [15]. Moreover, it is very safe (0.02 injuries per 1000 h in indoor climbing) [16,17,18,19]. Desirable psychological effects can also be produced. For example, climbers can enjoy the challenge of reaching extraordinary heights and develop a sense of responsibility for their partners and win their trust, improve concentration and develop cooperation and respect [20]. When climbing, one experiences intense feelings, such as fear, joy and pride. They view the competitiveness between climbers as less important compared with other antagonistic sports. All these aspects are related to individual autonomy and competence, which can be combined with activation and valence to enhance intrinsic motivation according to self-determination theory [20,21]. Thanks to its physical and psychological advantages, TC was a natural development. Research indicated that the earliest study on TC was conducted by Bienia, a Polish scholar. In his study, climbing was a factor in rehabilitation treatment, and TC was adopted in the field of orthopedics [22,23,24]. In the 1980s, it was applied to psychiatric disease intervention. Mcclung [25] was the first to treat chronic mental patients with TC. In his experiment, six patients suffering from schizophrenia or schizotypal personality disorder were treated with TC for 6 weeks. The results indicated that their physical and mental health improved greatly. However, TC at that time was still in its infancy, and scholars and medical specialists knew little about it. In the late 20th century, scholars paid more attention to it and adapted it to interventions with mentally handicapped children and drug addicts [26,27]. Since the beginning of the 21st century, with the boom in rock climbing and the severe situation regarding chronic diseases, TC has grown rapidly. It is increasingly used in occupational and physical therapy, as well as experiential education and psychotherapy, such as in the fields of orthopedics, neurology, geriatrics and psychotherapy [28]. Through TC, stroke and multiple sclerosis patients can become healthier, the elderly can improve their physical fitness and mobility, and victims of depression can release anxiety to improve their self-efficacy [29,30,31]. Nowadays, rock climbing has become an Olympic sport and is popular around the world. Therefore, more and more educators and therapists incorporate TC into their practical activities.

## 4. The Effects of TC

### 4.1. Depression

The World Health Organization reported that there were 322 million patients with depression worldwide, nearly half of whom lived in Southeast Asia and the Western Pacific (including India and China). From 2005 to 2015, the total number increased by 18.4%, and the disease burden of depression is expected to rise to first place in the world by 2030 [32]. In recent years, the therapeutic effects of exercise therapy were shown to be remarkable and were recognized by most scholars and the medical community [33,34]. However, many patients have poor physical health, a low fitness level and low physical self-worth. They are not motivated to perform strenuous physical activities and suffer from barriers to participation, such as physical and mental complaints and low self-confidence. As a result, they cannot withstand the treatment for a long time [34]. Studies showed that TC has a better impact on patients suffering from depression than traditional physical exercises and cognitive-behavioral group therapies [35,36]. Against this backdrop, scholars explored the effects of TC on patients with depression. For example, Luttenberger et al. [37] assessed its effects on 47 patients through a randomized cross-control experiment for 8 weeks. These patients practiced TC for 3 h each week. The BDI-II score rose by 6 points and their social skills, self-management skills, concentration, self-efficacy and anxiety relief were strengthened. Stelzer et al. [38] conducted a randomized controlled experiment where the participants practiced TC for 8 weeks. The experiment results revealed that the score for depression decreased by 6.74 points for the SCL-90-R and the score of BDI-II fell by 8.26 points, which is the normal level from the clinical view. Apart from these results, generally speaking, research suggested that TC has positive influences on reducing the level of depression [29,30,31], easing acute emotion, improving physical conditions and stabilizing mood [39,40,41]. These positive impacts can last for one year [42,43]. In a multi-center randomized controlled trial conducted by Karg et al. [44], 133 depression outpatients were divided into two groups of TC or home exercise for 10 weeks for an intervention. Compared with the home exercise group, the Montgomery Depression Rating Scale (MADRS) depression scores dropped more sharply for the TC group (drop of 8.4 vs. 3.0 points, *p* = 0.002). Moreover, the score variation for anxiety, body image and overall self-esteem also presented obvious differences between the two groups. Thus, TC is a promising method for treating depression. If someone has a systematic plan to practice it for a long time, their disease condition will improve [45,46,47]. The patient will feel better and can greatly enhance their social skills, self-management skills and self-efficacy.

### 4.2. Lower Back Pain

Low back pain refers to pain in any part of the human body between the costal margin and the gluteal fold, which may or may not be accompanied by symptoms of lower extremity discomfort [48]. It is a symptom syndrome represented by back pain. From 1990 to 2015, the years of disability due to low back pain increased by 54% globally, and it is the main reason for disability worldwide [49]. The European guidelines for the management of chronic nonspecific low back pain advise that exercise therapy can be an intervention [50]. Studies showed that TC can activate muscles, guide patients to view pain properly, and reduce pain-related fear and avoidance [51,52,53]. Patients can then develop a positive attitude to overcome their disease [54]. Dittrich et al. [55] performed an experiment to treat 55 patients with low back pain using TC for 3 months. The results suggested that the pain was eased, the body could move more flexibly and mental health was improved. In a randomized controlled experiment, the different impacts on chronic low back pain due to TC and standard exercise therapy were compared and assessed. All 28 patients were grouped into two teams, each treated with one of the two therapies four times a week for 4 weeks. There was no difference before and after treatment in terms of the Hanover Functional Ability Questionnaire, which is used for assessing low back pain. The two groups showed no marked improvement on three of the eight subscales of the 36-item Short-Form Health Survey (SF-36), which were vitality, mental health and social functioning. For the subscale of body role limitation, the standard exercise therapy group performed better. TC functioned better on two subscales (physical function and general health), indicating that the effect of TC was as good as and even partially better than the standard exercise [54,56]. Kim compared the effects of TC and lumbar stabilization exercise over 4 weeks on 30 adult patients with chronic low back pain. After the intervention, they were assessed using the 36-item Short-Form Health Survey (SF-36) and their psoas surface electromyography (sEMG) was checked [57]. It was found that the SF-36 scores rose dramatically in both groups, with a greater increase in the TC group, along with a sharp increase in the psoas sEMG activity in both groups. During the exercise, the sEMG activity of the rectus abdominis and the internal and external obliques increased more than the lumbar stabilization group. This suggested that TC, which is similar to normal lumbar stabilization exercises, can effectively activate and improve the function of the psoas muscles and the stability of the lumbar region. Meanwhile, Schinhan et al. [58] designed a prospective randomized controlled trial to compare the effects of TC on patients with chronic low back pain. The patients were divided into two groups, where one group practiced rock climbing while the other did not. The results revealed that the differences in the Oswestry disability index and visual analog scale (VAS) between the two groups were statistically significant (*p* = 0.022), and the area of intervertebral disc herniation in the TC group was significantly reduced. The above studies demonstrated that TC can improve various physiological functions and health conditions of patients with low back pain.

### 4.3. Multiple Sclerosis

Multiple sclerosis is an immune-mediated disease that is characterized by inflammatory demyelinating lesions of the central nervous system (CNS) [59]. The common symptoms are body sensory disorders, body movement disorders, ataxia and cognitive impairment. It is the most common non-traumatic disabling disease affecting young people [60]. Scientific research indicated that patients with multiple sclerosis should participate in more physical activities to control symptoms, recover body function, optimize quality of life and improve health [61,62]. However, because the patients are not as active as healthy people, they are not as eager to exercise. Despite this, TC works well to motivate patients to participate in the exercise and may have better outcomes than conventional therapy [23]. A randomized controlled trial was conducted to evaluate the effects of TC on motor activity and psycho-social factors in patients with multiple sclerosis. Twenty-seven patients took part in the experiment for 2 h a week for 6 months. The results indicated that TC can be considered a therapeutic medium for treating individual disabilities, enhancing independence and increasing the amount of activity. The fatigue value of the TC group decreased from 36 to 17.5 (*p* = 0.011), and there was no striking change in the controlled group [63]. Velikonja et al. [28] designed a randomized prospective study in which 20 patients with multiple sclerosis aged 26–50 years participated in 10 weeks of TC or yoga. The results found no significant effect on mood in both groups, with a 25% reduction in EDSSpyr in the TC group, but a 32.5% reduction in fatigue. There is also no significant effect on performance on selective attention and executive function tests, which can be proved by a 17% increase in performance on selective attention and no effect on executive function in the yoga group. Therefore, it is believed that TC can be used as a complementary way to relieve spasticity and fatigue. Kern [23] explored TC’s effects on multiple sclerosis through a randomized controlled trial. In the trial, 27 adults aged from 27 to 60 years practiced TC for 2 h per week for 6 months. The results indicated that the patients’ self-confidence, positive emotions, sense of balance and flexibility, and quality of life were significantly improved, and their fatigue was eased greatly. Subsequently, the patients were evaluated for the efficacy of TC at baseline, 6 months and 3 years. Overall, the patients significantly improved in terms of cognition, athletic ability and self-confidence. Their fatigue and depression were greatly relieved (*p* ≤ 0.001). Frederik [64] and Steimer et al. [65] also came to similar conclusions. It was found that the patients’ fatigue was significantly reduced, and their quality of life, health level and cognitive function were significantly increased after the therapeutic rock-climbing intervention. In conclusion, research showed that TC is beneficial to patients with multiple sclerosis by improving their physical fitness, easing fatigue, enhancing their self-efficacy and improving their quality of life.

### 4.4. Other Diseases

Many foreign studies provided new ideas for the rehabilitation of patients with diseases other than the abovementioned (Table 1). For example, TC positively affects the prevention and treatment of shoulder joint trauma [66]. It increases the range of motion of the shoulder muscles of patients with impingement syndrome and reduces their disability scores, which can be attributed to the fact that TC activates shoulder muscles, strengthening the muscles and coordination [67,68,69]. In addition, studies showed that TC can also be viewed as a means of rehabilitation after calcaneal fractures [70], a complementary treatment option for patients with scoliosis [71], and a way to enhance the self-worth and concentration of patients with anxiety and obsessive compulsive disorders [72,73]. It can reduce the annual bleeding rate and improve the joint condition of patients with hemophilia [74,75,76,77], promote self-esteem and self-efficacy for alcohol addicts, and improve adolescent gait function for children with cerebral palsy [78,79,80,81]. For children with attention deficit hyperactivity disorder, it can activate their brain waves and further focus their attention [82]. For patients with gynecological cancer, it can improve their health and functional suitability [83]. Moreover, it can improve speed up movement and strengthen the balance and flexibility of children with autism [84,85]. It can also improve the physical health of children with cancer and intellectual disabilities [86,87], intensify movement and coordination for patients with cerebellar ataxia, and improve the quality of life and upper limb function for stroke patients by enhancing their balance and walking function [88,89,90]. Furthermore, it can alleviate the symptoms of Parkinson’s patients by boosting their self-confidence and dynamic balance ability [91,92], and the static balance and gait can improve for patients with a spinal cord injury [93]. Overall, we found TC to be a potent treatment method for the improvement of physical fitness or disease rehabilitation of the abovementioned patients. However, the current literature only provides a basis for rehabilitation. In follow-up studies, the intervention effect should be further explored through high-quality randomized controlled trials.

### 4.5. Limitations of TC

Although there is ample evidence proving the benefits of TC for a variety of ailments, it must be specifically noted that exercise is rarely prescribed. It is reported that only one-third of people who consult a physician in the United States receive exercise counseling [94]. Although exercise habits are important predictors in exercise counseling, and current research suggests that experts have a positive attitude toward exercise therapy in general and TC in particular, this does not appear to be a decisive reason for prescribing exercise. For physicians, they considered a lack of time and reimbursement as barriers [20]. Although most exercise counseling studies have been conducted in the United States and other developed countries, the economic costs of TC (e.g., equipment, facilities) appear to be higher than standard exercise therapy in the current study. The financial aspect is often referred to as a side effect of prescription and TC [9]. There is one point that should be emphasized: TC is not suitable for patients with advanced osteoporosis, extreme obesity, incomplete healing of fractures, acute illness, acute pain, etc. Patients in the acute inflammatory phase, i.e., bilateral skin temperature contrast T > 2 °C or pain during training are considered contraindicated. In patients without any exercise experience, attention is given to guiding them to distinguish between the feeling of muscle soreness and training pain.

## 5. Potential Mechanisms of TC

### 5.1. Physiological Mechanisms

When practicing TC, static and dynamic muscles work alternatively. The dynamic muscle follows the process from dynamic to centripetal and then dynamic to centrifugal to strengthen the neuromuscular adaptability. Centripetal movement has a unique advantage in disease recovery since it can increase cortical excitability and decrease intracortical inhibition and spinal cord excitability to develop neuromuscular adaptability [95,96]. Furthermore, the entire treatment process is conducted in a so-called “closed chain”. Climbing can compress the joints without generating joint shearing forces, increasing the stability of the joints so that the stress on the limbs prevents muscle atrophy and osteoarthritis. Despite training being a closed chain, TC requires better coordination. There are two reasons for this, one is that a small rock support can allow for the entire body’s center of gravity to be balanced very well. The other is that the entire body moves diagonally during the climbing process. The complex diagonal movement is functional and is more effective than simple movements. Therefore, the climber must focus on a specific muscle or muscle group that involves the complete myofascial chain, which helps damaged or weakened structures readapt [8]. At the same time, the entire musculoskeletal system is improved by adopting a three-dimensional movement pattern in which pressure and stretch sensations in the muscles, as well as tendons, joint capsules and connective tissues, are stimulated by pressure, stretching and relaxation (it is similar to the PNF pattern) [8]. The relatively small support area of the hold and the demands on the limbs and core stability improve motor and postural control. As many studies showed, the diverse and equally targeted training of functional muscle groups and the entire musculoskeletal system in rehabilitated patients is effective.

### 5.2. Psychological Mechanisms

#### 5.2.1. Social Support Hypothesis

Social support means that an individual can feel, perceive or receive care or assistance from others. Many climbers can take part in TC at the same time and they must rely on each other’s protection. A climber frequently transfers their role between climber and protector to solve problems together and encourage and support each other, which develops the spirit of camaraderie and encouragement. Research showed that climbers can be less sensitive to interpersonal relationships and have more skills to cope with them [38]. Meanwhile, they are inspired not only by their achievements but also by the success of their peers. They are more willing to assist others to solve a variety of psychological problems and improve mental health.

#### 5.2.2. Distraction Hypothesis

Attention is the pointing and focusing of mental activity on a certain object and has both pointing and focusing characteristics. When an individual directs and focuses their awareness on a certain activity, their awareness of everything around them decreases accordingly. The distraction hypothesis was first conceptualized about 40 years ago [97], and distraction theory suggests that exercise can distract an individual’s attention from unfavorable stimuli, ignore negative stimuli and focus on positive stimuli, thus achieving an improvement in mood [98,99]. In other words, the cognitive resource for individual’s attention is limited, and when it is fully occupied, new stimuli are not processed. When the individual allocates it to important activities, for example, when one is addicted to exercise, negative emotions will be excluded from attention and the individual can only experience the positive feelings from exercise. The body’s contact area with the climbing wall remains small during therapeutic climbing, which ensures a high level of concentration. During the process of TC, the patient focuses on climbing movements to balance their body and adjust their breathing, which takes their attention away from pain and disability and focuses on individual energetic experience. In addition, the patient will change their idea on pain and not be too frightened to elude it. They will be more optimistic about improving their health instead of complaining about the disease. Then, their depression can be eased and the cognitive deficits and bias will disappear [40].

#### 5.2.3. Self-Efficacy Hypothesis

Self-efficacy is defined as a person’s confidence in their ability to make full use of their skills to complete a certain job. Four factors may enhance or weaken individual self-efficacy: successful experience, substitution experience, verbal persuasion and physiological condition, of which the most effective factor in developing self-efficacy is successful experiences [100]. In terms of TC, the exercise is often modular and can be performed flexibly. The modules vary in scope and complexity from single to complex motion sequences. In this way, the training can be flexibly designed so that the exercise can be quickly adapted to the current fitness level. We can grade the level of exercise elaborately based on height and difficulty, meeting the patient’s needs via tiny adjustments to achieve a sense of achievement. Through continuous rock climbing, the patient gains the joy of success, their strong positive emotion is activated and self-efficacy is enhanced. Thereby, their confidence in the face of disease is strengthened. Challenges are important when one takes positive action because the complex reward system of the human brain allows individuals to experience a special sense of well-being when challenges are successfully overcome. In addition, a therapeutic component that is inherent in current TC involves all aspects of mindfulness. With brief mindfulness exercises at the beginning and end of the practice, the patient can perceive their current physical and emotional condition during mindfulness meditation instead of assessing their current disease situation, thus diverting attention away from negative thoughts so that their depression and mood are improved [34].

## 6. Discussion

Current research suggests that TC is a very promising therapeutic tool, especially for the physical or psychological effects of depression [41,42,43,44], lower back pain [55,56,57,58] and multiple sclerosis patients [23,28,64,65]. However, there is a lack of empirical evidence on the effectiveness of rehabilitation for people with other diseases such as hemophilia and Parkinson’s, and the current study only provides one way of thinking about their rehabilitation. At this stage of the study, the following limitations exist in the field of therapeutic rock climbing. (1) There is no high-quality evidence to make definitive exercise recommendations. For example, most studies are non-blinded trials. (2) The data are not comprehensive enough, and some studies do not have a control group to demonstrate the superiority of TC over other therapeutic exercise interventions. (3) Many of the existing studies on TC in the literature do not reflect the potential negative effects such as economic aspects, possible overburdening of the patient, the deterrent effect of pain and injury, and the factor of untrained therapists. The therapist, in particular, is the weakest link in the therapeutic process and requires not only a high level of therapeutic knowledge but also a significant amount of rock-climbing expertise so that individualized programs can be assigned to patients with different illnesses. Therefore, in the future, exercise guidance programs for chronic disease prevention and treatment should be actively developed through large-sample, high-quality randomized controlled trials in terms of exercise frequency, exercise intensity, exercise duration, total exercise volume and progression, different ages and health level. TC interventions should be compared with other exercise modalities, such as stretching or aerobic exercise with psychotherapeutic intervention, to further explore the clinical effects of TC and the interaction effects with other therapies. With the aim of knowing more about the effects of TC, the following factors including adverse effects (e.g., economic aspects, excess burden, pain) and potential benefits (e.g., in psychological, social, and physiological domains) should be taken into account and evaluated with larger samples and controlled designs.

## 7. Conclusions

TC provides a new way of thinking about the rehabilitation and treatment of patients with depression, low back pain, hemophilia and other diseases. On the one hand, the movement pattern of muscles that are used when climbing is complex, diagonal and functional. The closed-chain movement pattern is beneficial to the patient’s neuromuscular and musculoskeletal system. On the other hand, TC is designed to be refined and modular. Its intensity has a significant effect on the patient’s self-efficacy. Because of the high concentration and social support when climbing, the patient improves their mental health and has more courage to face challenges.

## Figures and Tables

**Figure 1 ijerph-19-09696-f001:**
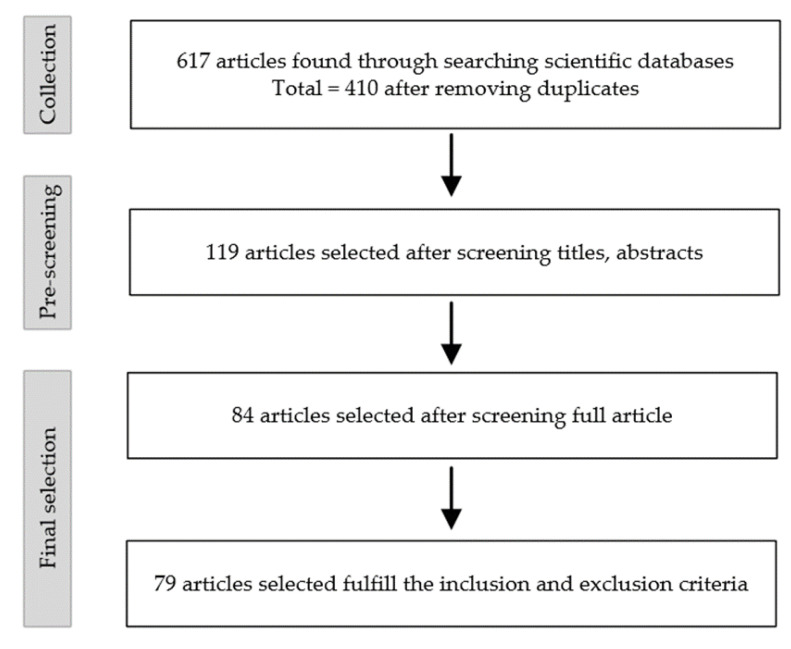
Overview of article selection process used in this narrative review.

**Table 1 ijerph-19-09696-t001:** Intervention effects of TC on patients with various diseases.

Study	Disease Type	N	Sex	Age (Years)	Study Design	Main Conclusions
Stemberger et al. [74]	Hemophilia	1	M	25	34 weeks TC	Bleeding rate ↓, joint health ↑, quality of life ↑, rock climbing ability ↑
Stemberger et al. [75]	Hemophilia	9	-	16–35	2 h × 34 weeks TC	Bleeding rate ↓, joint health ↑, quality of life ↑
Schroeder et al. [76]	Hemophilia	12	M	31	12 times TC	Range of motion ↑, rock climbing ability ↑, quality of life ↑, inflammation ↓
Schroeder et al. [77]	Hemophilia	6	M	≥21	12 times TC	Bleeding rate ↓, joint health and dorsiflexion ↑, quality of life ↑, rock climbing ability ↑
Lee et al. [82]	Attention deficit hyperactivity disorder	1	M	7	3 times × 4 weeks × 60 min TC	Brain waves and attention ↑
Crawford et al. [83]	Gynecologic cancer	E: 24	F	52.5 ± 12.7	2 times × 8 weeks × 2 h TC	the E is superior to the C for the 6 min walk, 30 s chair stand, 30 s arm curls, sit and reach, 8-foot up-and-go, grip strength-right, and grip strength-left assessments.
		C: 11	F	54.1 ± 10.5	2 times × 8 weeks × 2 h regular exercise	
Böhm et al. [80]	Cerebral palsy	E: 8	6M2F	13 ± 4.3	2 times × 6 weeks × 1.5 h TC and then traditional exercise	Walking speed ↑, step length ↑
		C: 8	6M2F	13 ± 4.3	2 times × 6 weeks × 1.5 h traditional exercise and then TC	Gait profile score ↑, ankle dorsiflexion ↑, knee flexion ↑, walking speed ↑, step length ↑
Christensen et al. [81]	Cerebral palsy	E:11	4M7F	11.6 ± 0.8	3 times × 3 weeks × 150 min TC	Climbing ability ↑, number of climbing routes ↑, sitting-standing test ↑, muscle coherence ↑,maximal hand or finger strength-,cognitive abilities or psychological well-being -
		C:6	4F2M	11.8 ± 0.9		Climbing ability ↑, Climbing speed↑, maximal hand or finger strength-, cognitive abilities or psychological well-being -
Kokaridas et al. [84]	Autism spectrum disorder	E:3	M	9	2 times × 12 weeks × 40 min TC	Grip strength ↑, speed ↑
		C:3	M	9		
Daggelmann et al. [86]	Cancer	13	5F8M	11.5 ± 4.47	1 time × 8 weeks × 60 min TC	Dorsiflexion strength ↑, Ankle dorsifleion-range of motion ↑, legs flexed ↑
Bibro et al. [87]	Intellectual disabilities	E: 32	13F19M	21.8 ± 2.5	2 times × 15 weeks × 60 min TC	Balance ↑, arm hang test↑, distance to push a 2 kg solid ball ↑, grip strength ↑
		C: 36	8F28M	19.8 ± 2	No intervention	No significant change
Taylor et al. [85]	Autism	7	M	8–14	1 time × 6 weeks × 90 min TC	Cognitive tracking test and grip strength ↑
Marianne et al. [88]	Cerebellar ataxia	4	M	22–56	6 weeks of TC	Speed ↑, balance ↑, hand flexibility ↑
Park et al. [89]	Stroke	E: 7	6M1F	45.43 ± 16.46	3 times × 6 weeks × 30 min TC	Compared with the control group, the experimental group showed quality of life ↑, upper limb function ↑, vitality ↑, mental health ↑
		C:7	5M2F	55.57 ± 7.39	Traditional treatment	
Lee et al. [90]	Stroke	E: 7	6M1F	45.43 ± 16.46	5 times × 60 min × 6 weeks standard rehabilitation exercise + 3 times 30 min × 6 weeks TC	Compared with the control group, the experimental group showed balance ability and walking ability ↑
		C: 7	5M2F	55.57 ± 7.39	5 times × 60 min × 6 weeks standard rehabilitation	
Kim et al. [67]	Shoulder impingement syndrome	E: 10	5M5F	54 ± 4.1	3 times × 8 weeks of TC	DASH score ↓, flexion and abduction ↑, external and internal rotation ↑, upper trapezius activity ↑
		C:10	6M4F	55.6 ± 7.4	8 weeks general isometric exercise	DASH score ↓, flexion and abduction ↑, serratus anterior and lower trapezius activity ↑
Woolstenhulme et al. [92]	Parkinson’s disease	3	M	70–73	3 × 8 weeks of TC	Confidence ↑, leg strength ↑, dynamic balance ↑
Telebuh et al. [93]	Spinal cord injury	1	M	34	1–2 times × 12 weeks × 90 min TC + 30 min home exercise	Static balance ↑, gait ↑
Langer et al. [91]	Parkinson’s disease	48	30M18F	64 ± 8	12 weeks TC vs. 12 weeks physical exercise	MDS-UPDRS III score ↓, symptoms of bradykinesia, rigidity and tremor ↓ vs. no significant change

These papers do not mention statistical significance [74,75,77,82,85,93], but others do. N—number of participants; M—male; F—female; E—experimental group; C—control group; study design—the number of times per week × duration + training content; ↑ = Significant increase; ↓ = Significant decrease; - = No significant change; EDSS—Expanded Disability Status Scale; DASH—disabilities of the arm, shoulder and hand; MDS-UPDRS III—Parkinson’s Disease Comprehensive Rating Scale Part III.

## Data Availability

Not applicable.

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
