# Peer review of "The Origin, Application and Mechanism of Therapeutic Climbing: A Narrative Review"

_ijerph, 2022, doi:10.3390/ijerph19159696_

Round 1

Reviewer 1 Report

Congratulations to the authors of this interesting manuscript. Below I present suggestions to increase the quality and readability of the work.

Introduction:

- at the end of the introduction, a clearly formulated aim of the work should appear

- the methodology of searching and selecting works for publication should be described in a precise and legible way - how the authors found works from the given databases. Why re-screening was performed. It is worth presenting this as a flow diagram

- there should be information about what the TC is, with what intensity the effort should be performed, with what frequency. Who is it for / contraindicated for. The authors should move some of the text from page 9 (However .... (...)) to accurately describe who should apply this training.

The main part of the job.

The authors wanted to do a narrative review. It should be critical in terms of a given topic. I get the impression that the work is written very biased. It does not include information on the difficulties in using TC in the diseases described by the authors. We have no negative work related to this kind of exercise. Are there no such works?

What's more, the sentence: TC is better than other kinds of sports and there is evidence showing that it is a helpful intervention. It's worded very strongly. Are you sure - no other type of exercise is recommended?

Table 1: I suggest to divide the table to the sections:  depending on age, gender and diseases - to be more clear.

Reviewer 2 Report

Dear authors, it is an interesting article however, 

The main purpose of a narrative review is to deepen the understanding in a certain research area and the appraisal of previous studies conducted on a certain topic. 

You conclude that therapeutic climbing “provides a new means for the treatment of depression, lower back pain, multiple sclerosis and other diseases, and its treatment effect is better than traditional exercise treatments.” Yet your review doesn’t provide any evidence for that.

The major problem of your review is that you completely ignore all the problems you use in your every review. These input these problems include, Nonblinded studies that rely on subjective outcomes. The studies are small small or very small. Even though it’s known that the subjective outcomes are unreliable in Nonblinded studies and that subjective outcomes are unreliable in exercise studies, both in healthy and the ill, almost none of your studies uses objective outcomes. Most studies also do not use a control group or a properly designed control group.

Please separate methods from introduction.

In table one you mention the studies you examined yet a number of studies you discuss in your article are not mentioned in it, for example, Velikonja et al..

All the climbing studies you discuss in your article should also be in that table.

I didn’t check all the studies you mention, however I did check Velikonja et al..

You state the following about that study.

“Velikonja et al. [24] observed a 25% reduction in the Extended Disability Status Scale and a 32.5% reduction in fatigue during a 10-week TC intervention involving patients with multiple sclerosis. They also explored TC’s effects on multiple sclerosis through a randomized controlled trial. In the trial, 27 adults aged from 27 to 60 years practiced TC for 2 hours per week for 6 months. The results indicated that the patients’ self-confidence, positive emotions, sense of balance and flexibility, and quality of life were significantly improved, and their fatigue was eased greatly. Subsequently, the patients were evaluated for the efficacy of TC at baseline, 6 months and 3 years. Overall, the patients significantly improved in terms of cognition, athletic ability and self-confidence. Their fatigue and depression were greatly relieved (p ≤ 0.001).”

However, according to Velikonja et al. themselves it’s not a study with 27 adults and they also reach different conclusions is highlighted by the following from their study:

“20 subjects with relapsing–remitting or progressive MS, 26–50 years of age, with EDSS ≤ 6 and EDSS pyramidal functions score (EDSSpyr) > 2 were enrolled in a randomized prospective study. The participants were randomly divided into SC and yoga group.”

“There were no significant improvements in spasticity after SC and yoga. In the SC group we found a 25% reduction (p = 0.046) in EDSSpyr. There were no differences in executive function after the completion of both programs. There was a 17% increase in selective attention performance after yoga (p = 0.005). SC reduced fatigue for 32.5% (p = 0.015), while yoga had no effect. We found no significant impact of SC and yoga on mood.”

So please carefully check all the studies again. 

Also please pay more attention to the weaknesses of those studies and pay attention to the objective outcomes. 

Reviewer 3 Report

I thank the Editor for the opportunity to serve as Reviewer for this interesting and innovative manuscript. The article is well written, however I suggest some major changes before it can be accepted. 

First, I suggest making the abstract more impersonal, specifying when referring to we who should create the therapeutic climbing protocols. 

I would add a reference to line 26 of the introduction.

In the introduction I suggest adding references to other manuscripts that have dealt with alternative methods of rehabilitation (I suggest mentioning for example Maccarone, M.C., Magro, G., Solimene, U. et al.)

1. From in vitro research to real life studies: an extensive narrative review of the effects of balneotherapy on human immune response. Sport Sci Health 17, 817–835 (2021). https://doi.org/10.1007/s11332-021-00778-z; Masiero, S., Maccarone, M.C.

 2. Demers M, Thomas A, Wittich W, McKinley P. Implementing a novel dance intervention in rehabilitation: perceived barriers and facilitators. Disabil Rehabil. 2015;37(12):1066-72. doi: 10.3109/09638288.2014.955135. Epub 2014 Aug 28. PMID: 25163831.

3. Pereira APS, Marinho V, Gupta D, Magalhães F, Ayres C, Teixeira S. Music Therapy and Dance as Gait Rehabilitation in Patients With Parkinson Disease: A Review of Evidence. J Geriatr Psychiatry Neurol. 2019 Jan;32(1):49-56. doi: 10.1177/0891988718819858. Epub 2018 Dec 17. PMID: 30558462.

4. Health resort therapy interventions in the COVID-19 pandemic era: what next?. Int J Biometeorol 65, 1995–1997 (2021). https://doi.org/10.1007/s00484-021-02134-9; Tognolo, L.; Coraci, D.; Fioravanti, A.; Tenti, S.; Scanu, A.; Magro, G.; Maccarone, M.C.; Masiero, S.

5. Clinical Impact of Balneotherapy and Therapeutic Exercise in Rheumatic Diseases: A Lexical Analysis and Scoping Review. Appl. Sci. 202212, 7379. https://doi.org/10.3390/app12157379)

I suggest removing the indications of materials and methods (databases consulted, studies selected) from the introduction and creating a separate section. 

Please further summarize the results, so as to make them more easily accessible for the busy reader. I also suggest moving lines 209-212 to a discussion section.

In Table 1, I suggest specifying whether the results obtained are statistically significant and which outcome measures were taken into account.

Section 4.2.2. Distraction Hypothesis should be expanded by adding more references to support what is written.

I suggest backing up the interesting statements in the conclusions with more references in the text (e.g. by adding a separate section on contraindications). 

Round 2

Reviewer 2 Report

Dear authors, Thank you for making the changes

Reviewer 3 Report

I read with interest the revised version that can now be accepted. Thank you very much for your corrections.